# Association of Intracellular Microstructural and Neuropsychological Changes in HIV: A Pilot Validation of Trace Diffusion-Weighted Magnetic Resonance Spectroscopic Imaging Using Radial Trajectories

**DOI:** 10.3390/metabo15100669

**Published:** 2025-10-13

**Authors:** Ajin Joy, Andres Saucedo, Matthew J. Wright, Pranathi Vallabhu, Neha Gupta, James Sayre, Aichi Chien, Uzay Emir, Paul M. Macey, Eric S. Daar, M. Albert Thomas

**Affiliations:** 1Radiological Sciences, David Geffen School of Medicine, University of California Los Angeles, Los Angeles, CA 90095, USA; ajinjoy@mednet.ucla.edu (A.J.); saucedo.and@gmail.com (A.S.); vallabhupranathi@gmail.com (P.V.); ngupta19097@gmail.com (N.G.); jsayre@mednet.ucla.edu (J.S.); achien@ucla.edu (A.C.); 2Physics and Biology in Medicine IDP, BioEngineering, David Geffen School of Medicine, University of California Los Angeles, Los Angeles, CA 90095, USA; 3Psychology, California State University, Fresno, CA 93740, USA; mattwright@mail.fresnostate.edu; 4Psychiatry, University of California, San Francisco, CA 94158, USA; 5Radiology, School of Medicine, University of North Carolina, Chapel Hill, NC 27514, USA; uzay_emir@med.unc.edu; 6School of Nursing, University of California Los Angeles, Los Angeles, CA 90095, USA; pmacey@sonnet.ucla.edu; 7Medicine, David Geffen School of Medicine, University of California Los Angeles, Los Angeles, CA 90095, USA; edaar@lundquist.org; 8Lundquist Institute at Harbor-UCLA Medical Center, Torrance, CA 90502, USA

**Keywords:** magnetic resonance imaging, HIV, diffusion-weighted imaging, choline, creatine, N-acetylaspartate, anxiety, depression, attention, executive function, memory

## Abstract

**Background:** Despite effective antiretroviral therapy, HIV-associated neurocognitive disorders (HANDs) remain prevalent, highlighting the need for sensitive biomarkers of early brain alterations. Trace-weighted diffusion spectroscopic imaging offers a non-invasive means to assess microstructural changes in brain metabolites in a single shot by measuring apparent diffusion coefficients (ADCs) of total N-acetylaspartate (tNAA), total creatine (tCr), total choline (tCho), and water. **Methods:** In this study, we used trace-weighted single-shot diffusion-weighted radial echo-planar spectroscopic imaging (DW-RESPI) to investigate metabolite diffusion and relative concentrations in the brains of people living with HIV (PLWH). Using a 3T MRI scanner, we studied 16 PLWH and 15 healthy controls (HCs), and we collected two sets of data with low and high b-values from which metabolite ADCs were computed. Metabolite ratios were derived from the low b-value spectra. A brief neuropsychological assessment evaluated attention, executive function, and memory in a subset of subjects. Cognitive and affective performance was quantified using domain-specific deficit scores, as well as depression and anxiety assessments, offering a comprehensive evaluation of neurobehavioral function. In the male subgroup (N = 15) of PLWH, we calculated the correlations between ADC values and neuropsychological domain scores. **Results:** tNAA, tCr, tCho, and water ADC values were significantly elevated in multiple gray and white matter regions in PLWH compared to HC, with the most pronounced differences observed in the superior precuneus, anterior cingulate cortex, and corona radiata. Notably, regional ADC values and metabolite ratios showed significant correlations with neuropsychological domain scores. **Conclusions:** These findings indicate the potential of metabolite and water diffusion metrics as biomarkers for HIV-associated microstructural brain alterations and cognitive impairment. However, the small sample size and preliminary nature of this data warrant further investigation to validate these findings.

## 1. Introduction

Quantification of brain structure and function through MRI is an important tool for studying human immunodeficiency virus (HIV) infection [1]. HIV-associated neurocognitive disorders (HANDs) range from mild cognitive impairment to frank dementia. Despite the availability of antiretroviral therapies (ARTs), these disorders continue to affect individuals with HIV [1,2,3,4,5], and HAND continues to affect many people living with HIV (PLWH). Recent cohort studies estimate the global prevalence of HAND anywhere from 25% to 60%, with variation due to differences in diagnostic criteria, study populations, and geographic settings [6]. The most common subtype, asymptomatic neurocognitive impairment (ANI), accounts for about 28%, followed by mild neurocognitive disorder (MND), while HIV-associated dementia (HAD) has become relatively rare in the ART era [7]. Although effective ART has significantly reduced the incidence of HAD, the overall prevalence of HAND has remained stable or even increased in some populations due to prolonged survival and aging of PLWH [8]. Risk factors contributing to HAND include older age, lower education, comorbid conditions, and duration of HIV infection [9]. HAND is associated with impairments in executive function, memory, and attention that reduce quality of life and treatment adherence. Consequently, identifying biomarkers of neuropsychological problems could help understand and address this persistent neurological complication in PLWH [10].

Traditional imaging modalities, such as structural and functional MRI techniques, provide some information about brain anatomy and activity, although they are limited in their ability to probe the biochemical and microstructural alterations that may accompany HIV infection [11]. Diffusion-weighted imaging (DWI), in particular, can be a powerful technique for studying the microstructure of the brain. DWI captures the movement of water molecules within tissues, which can provide insights into cellular integrity and tissue organization [11,12,13,14,15,16].

Recent advances in diffusion-weighted radial echo-planar spectroscopic imaging (DW-REPSI) combine the strengths of DWI and spectroscopic techniques, integrating two spatial dimensions with one spectral dimension, offering enhanced sensitivity to changes in tissue metabolism [17,18,19]. The approach allows assessment of tissue diffusion properties across different regions of the brain, along with the simultaneous characterization of biochemical profiles, providing a more comprehensive view of the brain’s physiological state.

In the context of HIV, the ability to distinguish pathological from healthy brain tissues using single-shot trace DW-REPSI could offer new insights into the mechanisms underlying HIV-associated neurocognitive impairment. HIV infection can lead to alterations in white matter (WM) integrity, cortical thinning, and regional metabolic changes [20,21,22,23,24,25,26,27,28]. Metabolite ratio and apparent diffusion coefficient (ADC) measurements from magnetic resonance spectroscopy (MRS) provide complementary information on brain tissue biochemical and microstructural properties affected by HIV infection. Key metabolites quantified by MRS include total N-acetylaspartate (tNAA), an established marker of neuronal integrity; total creatine (tCr), involved in cellular energy metabolism; and total choline (tCho), reflecting membrane turnover and inflammatory activity. Prior research has demonstrated alterations in these metabolites in PLWH, consistent with neuroinflammatory processes, glial activation, and neuronal injury [29,30]. The ADC metric captures the diffusivity of metabolites and water within tissues, which can be altered by microstructural changes such as cellular swelling, loss of tissue barriers, or inflammatory edema. For example, increased ADC values of creatine have been reported in conditions of hypermetabolism and neuroinflammation, suggesting altered energy metabolism and microglial activation underlying neuropathology [15,31]. The integration of MRS metabolic profiling with diffusion-based ADC measures enables a more sensitive and comprehensive characterization of HIV-associated brain alterations, with potential applications as early biomarkers of neurocognitive impairment and for monitoring treatment effects.

By exploiting diffusion characteristics of intracellular metabolites, this pilot study seeks to investigate whether there are significant differences in the metabolic signatures of the brain (total N-acetylaspartate (tNAA), total creatine (tCr), and total choline (tCho)) in PLWH compared to healthy controls (HCs). Specifically, an advanced spectroscopic imaging technique called trace-weighted single-shot diffusion spectroscopic imaging [19,32,33] is used in this study, which helps avoid unwanted effects from eddy currents and additional diffusion weighting from b-value cross-terms between imaging and diffusion gradients. Eddy current artifacts arise from rapid switching of gradient fields and cause frequency shifts and/or baseline distortions in the spectra [34]. Cross-term interactions, particularly those between the diffusion gradients and the imaging gradients used for localization, can cause errors in the measured apparent diffusion coefficients of metabolites [19,32,34]. By isolating the trace of the diffusion tensor, this technique provides a scalar measure of diffusion that is tissue structures orientation independent. This is particularly valuable in tissues with complex or unknown orientations.

This pilot study primarily seeks to apply trace DW-REPSI to detect brain changes in the PLWH that are not observable with conventional imaging, such as changes in metabolite ADCs. A secondary aim is to examine whether metabolic profiles differ between PLWH and HC, and whether these differences correlate with cognitive dysfunction, positioning DW-REPSI as a potential source of biomarkers for cognitive impairment and for identifying individuals at greater risk of cognitive decline.

## 2. Materials and Methods

### 2.1. Trace-Weighted Single-Shot Diffusion Spectroscopic Imaging

This advanced diffusion-weighted spectroscopic imaging technique is achieved by arranging 12 pairs of bipolar diffusion-sensitizing gradients (DSGs) along three spatial dimensions within a Point RESolved Spectroscopy (PRESS) sequence [19]. The particular configuration of DSGs, as outlined in [19,32], results in the cancelation of signals associated with the off-diagonal elements of the diffusion tensor. In addition, the double refocusing scheme in the pulse sequence helps to (a) reverse the phase accumulation due to static field inhomogeneities, (b) reduce eddy current-induced signal variations, and (c) eliminate cross-terms contributions between the diffusion-sensitizing gradients and static background gradients, thereby nullifying this contribution to the overall diffusion weighting [19,32,35].

### 2.2. Subjects

Sixteen HIV-infected participants (mean ± SD; 52.5 ± 10.5 years) and fifteen age-matched HIV-uninfected healthy subjects (mean ± SD; 37.6 ± 12.6 years) were recruited for this study from 2021 to 2024. Written informed consent was obtained from all participants, and the study participants were recruited from the Lundquist Institute at Harbor-UCLA Medical Center (Torrance, CA, USA), David Geffen School of Medicine at UCLA (Los Angeles, CA, USA), and clinicaltrials.gov. The study received ethical approval from the Institutional Review Board at both the University of California, Los Angeles, and the Lundquist Institute at Harbor-UCLA Medical Center. Participants were screened and excluded if they had a history or current diagnosis of alcohol or substance use disorders, attention deficit disorder (past or present), active depression or other psychiatric conditions, metabolic imbalances, metal implants, claustrophobia, pregnancy, or neurological disorders unrelated to HIV.

### 2.3. Data Acquisition

The brain imaging was conducted using a 3.0-Tesla Prisma-Fit MRI scanner (Siemens, Magnetom, Erlangen, Germany) equipped with a 16-channel head receive coil. High-resolution 3D T1-weighted structural images were obtained using a magnetization-prepared rapid gradient echo (MP-RAGE) sequence with the following parameters: repetition time (TR) = 2200 ms, echo time (TE) = 2.41 ms, inversion time (TI) = 900 ms, flip angle = 9°, matrix dimensions = 320 × 320, field of view (FOV) = 230 mm × 230 mm, slice thickness = 0.9 mm, pixel bandwidth = 200 Hz, and a total of 192 slices were acquired. Following the 3D MP-RAGE, the diffusion-weighted single-trace DW-REPSI data were acquired using the following parameters. FOV of 320 × 320 × 20 mm^3^ and matrix size of 32 × 32 × 1, resulting in a voxel volume of 2 mL. A VOI of approx. 75 mm × 115 mm × 20 mm was localized within the FOV. The TR/TE were 2250/144 ms. The radial echo-planar readout gradient sampled 512 time points with a spectral width of ~1190 Hz. Two b-values were measured: a low-value at 51 s/mm^2^ and a high b-value at 1601 s/mm^2^, achieved with DSG amplitudes of 11 mT/m and 64 mT/m, respectively. A non-water-suppressed scan was acquired for eddy current phase correction and coil combination. While six and fourteen averages were acquired for water-suppressed scans, one and two averages were acquired for non-water-suppressed scans of low b-value and high b-value, respectively. The total scan time, including both the water and non-water suppressed scans, was approximately 45 min.

### 2.4. Data Reconstruction and Post Processing

The DW-REPSI data were reconstructed using nuFFT [36] after gradient delay calibration [37] and self-navigation corrections [38,39,40]. Water reference data were used for eddy current phase correction [41]. Residual water was removed using Hankel–Lanczos singular value decomposition method [42], and the resultant spectra were quantified using LCModel 6.2-1L [43]. Only the concentration estimates with Cramer-Rao lower bounds (CRLBs) ≤20% were considered. The gray matter (GM) and WM fractions were calculated based on the MP-RAGE images, and the ADC values were computed as ADC_m_ = −log [S_m_(b_high_)/S_m_(b_low_)] (b_high_ − b_low_), where m denotes the metabolite.

The regions analyzed in this study include right and left anterior cingulate cortex (RACC and LACC), right and left superior precuneus (RSP and LSP), and bilateral anterior and posterior corona radiata (RACR (right anterior corona radiata), RPCR (right posterior corona radiata), LACR (left anterior corona radiata), and LPCR (left posterior corona radiata)). These regions have well-established involvement in HIV-associated neuropathology and cognitive dysfunction. The anterior cingulate cortex (ACC) is crucial for executive function, attention, and emotional regulation, with numerous studies linking ACC structural and functional alterations to HIV-related cognitive impairment [7,44]. The precuneus, a key hub in the default mode network, is implicated in memory, visuospatial processing, and self-referential thought and is frequently affected in HIV, with neuroimaging consistently showing metabolic and connectivity disruptions correlated with cognitive deficits [45]. The corona radiata, comprising essential white matter tracts, facilitates cortical–subcortical communication vital for executive functions, and the diffusion abnormalities in these regions are strongly associated with HAND and cognitive decline [46].

Since neuropsychological data were collected for only six HC, we restricted correlation analysis to the PLWH to avoid biased or uninterpretable findings due to the small control sample. This approach aligns with best practices in HIV neurocognitive research, where limited control data necessitate focusing on correlation assessments within PLWH [47,48].

### 2.5. Neuropsychological Testing

Sixteen HIV-infected and six HC participants completed a brief battery of neuropsychological tests. Given that our data collection occurred at the tail end of the COVID-19 pandemic, the neuropsychological test battery was completed over video teleconferencing. Earlier research reports that have shown that remote assessment with a variety of neuropsychological tests, including those used in this study, are reliable, valid, and well tolerated by participants [49,50,51]. For the present analysis, the neuropsychological tests included the Auditory Consonant Trigrams (ACTs; a test of executive ability) [52], Digit Span from the Wechsler Adult Intelligence Scale, third edition (WAIS III Digit Span; a measure of attention) [53], the Hospital Anxiety and Depression Scale (HADS; a measure of mood symptoms), the Lawton and Brody Instrumental Activities of Daily Living Scale (IADLS; a functional outcome measure), the National Adult Reading Test (NART; a measure of premorbid intelligence) [54], the Oral Trail Making Test (OTMT; measure of attention/processing speed and executive ability) [55], and the Rey Auditory Verbal Learning Test (verbal learning and memory). All tests were scored and normed per convention. Domain scores for cognitive domains were calculated based on theoretical grounds (i.e., based on the abilities assessed by each test) as well as significant correlation between said T-scores for each test resulting in an attention domain score comprising the WAIS III Digit Span (OTMT trial A did not correlate with the Digit Span), an executive domain score comprising the average of OTMT trial B and the ACT total score, and the memory score comprising the average of RAVLT trials 1–5 (learning), trial 6 (short delay recall), and trial 7 (long delay recall). Additionally, beyond cognitive domain T-scores, we also generated deficit scores for each cognitive domain and global deficit score (GDS), representing the average of the domain deficit scores [56].

### 2.6. Statistical Analysis

Statistical analyses were performed to assess group differences, associations with neuropsychological performance, and classification accuracy in distinguishing PLWH from HC. A Mann–Whitney U test was employed to compare age distributions between PLWH and HC groups [57]. The test showed no significant difference in age (*p* < 0.05), supporting comparability of groups on this measure. To minimize confounding from uneven sex distribution, the single female participant in the PLWH group was excluded from correlation analyses, and, additionally, three female participants from the HC group were excluded from metabolite ratio and ADC comparisons [58]. This ensured all analyses were performed in sex-matched male subgroups, thus reducing bias arising from sex-related metabolic differences.

Group Comparisons: To compare metabolite ratios and ADC values between the PLWH and HC groups, independent samples Student’s t-tests were used. Separate t-tests were conducted for each metabolite ratio and ADC measure in both GM and WM regions. Statistical significance was set at *p* < 0.05.

Correlation Analysis: Pearson correlation coefficients were calculated to assess the relationships between metabolite ratios or ADC values and neuropsychological test scores across all subjects. This analysis was performed separately for each metabolite ratio and ADC measure to identify potential associations with cognitive performance.

Discriminant Analysis: To evaluate the ability of imaging biomarkers to classify PLWH and HC subjects, linear discriminant analysis (LDA) was performed. Two classification models were constructed: the first model included metabolite ratios and water ADC values as predictors; the second model included both metabolite ratios and metabolite ADCs.

To evaluate the predictive value of various metabolite combinations, Fisher’s stepwise linear discriminant analysis (LDA) was employed as the multivariate method. Model performance was further assessed using receiver operating characteristic (ROC) analysis, with the area under the curve (AUC) serving as the primary metric of accuracy. All statistical analyses were carried out using IBM SPSS Statistics for Windows, Version 24.0 (IBM Corp., Armonk, NY, USA).

## 3. Results

Demographic and clinical details for both the PLWH and HC are presented in Table 1. CD4+ T-cell counts and plasma HIV RNA levels were measured at the time of assessment.

### 3.1. ADCs at Selected Voxels in GM and WM Regions

ADC values were computed from eight regions for three metabolites (tNAA, tCr, and tCho) and water, and compared between PLWH and HC groups. Figure 1 shows representative localizer images (sagittal, axial, and coronal), an NAA metabolite map, and an extracted spectrum (2 cm^3^) from a 63-year-old male participant with HIV (CD4 = 420 cells/µL, viral count < 20 copies/mL). It may be noted that the relatively higher noise level in the low b-value spectrum is due to the lower number of averages compared to the high b-value spectrum. Multi-voxel plot from the VOI (white box in the axial localizer image) is shown in Figure 2. Eight different regions used for the analysis are marked in green boxes. Figure 3 shows another example of an axial localizer image and a multivoxel plot from a 38-year-old HC. The LC-Model fit of an extracted spectrum is shown in the bottom right panel.

#### 3.1.1. Metabolite ADC Comparison of PLWH vs. HC Groups

In the comparison between the PLWH and HC cohorts, statistically significant changes in ADC values were observed for tNAA, tCr, and water. tNAA showed significant elevation in RSP (*p* = 0.0148), while tCr was significantly elevated in the RACC region (*p* = 0.0184). Water ADC values also showed significant differences at RACC (*p* = 0.0000) and LACC (*p* = 0.0459). In all cases, the ADC values were observed to be higher in PLWH compared to HC. ADC of water at RACC also remained significant after Bonferroni correction. Figure 4 illustrates the ADC values of metabolites and water across different brain regions for both HC and PLWH groups.

#### 3.1.2. Estimates of ADC in Pure WM and GM

The ADC values from individual voxels were extrapolated to find pure ADC values in gray and white matter, as shown elsewhere [59,60,61]. tNAA, tCr, and tCho showed increased ADC values in WM compared to GM, while water ADCs were higher in the GM compared to WM. This was the same in both PLWH and HC. Although the difference in water ADC was not statistically significant, significant differences were observed for tNAA and tCr in both the PLWH and HC cohorts. ADC values of tCho were significantly different in PLWH but not in HC. However, there were no significant differences between PLWH and HC for metabolites as well as water. A bar chart comparing the extrapolated ADC values of GM and WM in HC and PLWH is shown in Figure 5.

#### 3.1.3. Correlation with Neuropsychological Scores and Emotional Functions

Here we examined the correlation between regional ADC values and neuropsychological domain T-scores (attention, executive function (EF), memory, anxiety, depression, and GDS), reflecting overall neurocognitive dysfunction scores, and reflecting symptoms of anxiety and depression. This analysis showed that ADC of tNAA at RACC had a statistically significant moderate correlation with the EF scores (r = 0.69, *p* = 0.04). tCr and tCho also showed significant correlations at RACC, while tCr showed a moderate negative correlation with memory scores (r = −0.65, *p* = 0.03), tCho showed a moderate correlation with GDS (r = 0.65, *p* = 0.03), and a moderate negative correlation with depression (r = −0.63, *p* = 0.04). In addition, tCr showed moderate correlation with EF at LPCR (r = 0.71, *p* = 0.03) and with anxiety at RSP (r = 0.62, *p* = 0.04). Scatter plots of metabolite ADCs that showed significant correlations with neuropsychological domain scores are shown in Figure 6.

### 3.2. Metabolite Ratios at Selected Voxels in GM and WM Regions

tCr has been reported to vary with the HIV infection [29]. Therefore, metabolite ratios with respect to the sum of tNAA, tCr, and tCho were computed from the low b-value data [62]. Multiple regions with statistically significant differences in the metabolite ratios were found in both gray and white matter regions.

#### 3.2.1. Metabolite Ratio Comparison of PLWH vs. HC

Between PLWH and HC cohorts, statistically significant changes in metabolite ratios were observed for tNAA, tCr, and tCho. The tNAA ratio changes were significant in RACC (*p* = 0.0176), RACR (*p* = 0.0000), and RPCR (*p* = 0.0037). Similarly, the tCr ratio showed significant changes in the RACR (*p* = 0.0027) and RPCR (*p* = 0.0263), and the ratio of tCho at RACC (*p* = 0.0084) and RACR (*p* = 0.0034). While the tNAA ratios were reduced in PLWH compared to HC, the tCr and tCho ratios were elevated in PLWH compared to HC. A bar chart comparing the metabolite ratios at different brain regions between HC and PLWH is shown in Figure 7. tNAA at RACR and RPCR, tCr at RACR, and tCho at RACC and RACR remained significant after Bonferroni correction.

#### 3.2.2. Correlation with Neuropsychological and Emotional Function

Similarly to the correlation analysis of metabolite ADCs, correlations between metabolite ratios and neuropsychological domain T-scores, the GDS, and anxiety and depression scores were computed. Scatter plots of metabolite ratios that showed significant correlations with the domain scores are shown in Figure 8. Regarding general neurocognitive dysfunction, tNAA and tCr ratios significantly correlated with the GDS (at RACC, r = −0.64, *p* = 0.03 and at LSP, r = −0.73, *p* = 0.01), respectively, while tCho ratios correlated with memory (at RSP, r = 0.66, *p* = 0.27). With regard to mood, tCho ratios showed correlations with greater symptoms of anxiety scores (at RPCR, r = 0.77, *p* = 0.006) as did tNAA ratios (at RACR, r = −0.65, *p* = 0.03).

#### 3.2.3. Linear Discriminant Analysis

LDA for PLWH and HC groups based on the ratios of metabolites and water ADCs yielded a statistically significant discriminant model using tCr ADC (Wilks’s Λ = 0.819, *F*(1, 28) = 6.175, *p* = 0.019) and water ADC (Wilks’s Λ = 0.453, *F*(1, 28) = 33.783, *p* < 0.001) with an AUC of 0.975 (95% CI: 0.923–1). The corresponding ROC curve is shown in Figure 9A. In cross-validation, the discriminant analysis resulted in a sensitivity of 87.5% (95% CI: 73.0–100%) and a specificity of 93.3% (95% CI: 78.6–100%). The positive predictive value was 93.8% (95% CI: 74.7–99.7%), and the negative predictive value was 93.3% (95% CI: 74.7–99.7%). The cross-validated classification accuracy was 90.3%, providing a more robust estimate of model performance and reducing the risk of overfitting. The prevalence of the condition in the sample was estimated at 51.6% (95% CI: 33.4–69.4%).

Using the second model with both metabolite ratios and ADCs at the classification parameters, the LDA for PLWH and HC groups yielded a statistically significant discriminant model using tCr ratio (Wilks’s Λ = 0.864, *F*(1, 28) = 4.397, *p* = 0.045) and tCr ADC (Wilks’s Λ = 0.819, *F*(1, 28) = 6.175, *p* = 0.019), with an AUC of 0.854 (95% CI: 0.708–1). The ROC curve is shown in Figure 9B. In cross-validation, the discriminant analysis resulted in a sensitivity of 87.5% (95% CI: 60.4–97.8%) and a specificity of 73.3% (95% CI: 51.4–89.5%). The positive predictive value was 82.4% (95% CI: 55.8–95.3%), and the negative predictive value was 85.7% (95% CI: 56.2–97.5%). The cross-validated classification accuracy was 80.6%, providing a more robust estimate of model performance and reducing the risk of overfitting. The prevalence of the condition in the sample was estimated at 51.6% (95% CI: 33.4–69.4%).

## 4. Discussion

This study evaluated regional metabolite and water ADC values, as well as metabolite ratios, in both GM and WM among PLWH compared to HC using diffusion trace-weighted REPSI. The observed increases in ADC values for tNAA, tCr, tCho, and water in PLWH, in regions such as the precuneus and anterior cingulate cortex, suggest microstructural alterations associated with HIV infection. These findings are consistent with previous reports of neuroinflammatory changes and axonal injury in PLWH, which may contribute to increased diffusivity in affected brain regions [63,64]. No significant changes in tCho ADC values were found between groups.

Successful pilot validation of the diffusion trace-weighted REPSI technology in PLWH and HC demonstrates clearly the following: (1) the trace ADC can be described as a robust, more repeatable, diffusion metric; (2) this sequence measures the trace ADC (also known as. mean diffusivity—MD) of the major metabolites in a single TR (instead of in three separate measurements); (3) the trace ADC is more robust across different measurements where the patient set-up and orientation relative to the scanner may vary, because the trace ADC is a rotationally invariant quantity.

Comparisons between GM and WM further revealed a consistent pattern of higher ADC values for metabolites in the WM, while water ADCs were higher in GM, which is consistent with previous reports [65,66,67]. These differences are attributed to distinct microstructural properties of GM and WM, as well as the differential impact of diffusion time on the ADCs of different molecules [66]. However, these tissue patterns were not specific to HIV infection, as there were no significant group differences in ADC values when extrapolated to pure tissue compartments. The trends are likely due to intrinsic structural and biochemical properties of GM and WM, not HIV-specific pathology.

Although no significant differences were observed in the extrapolated metabolite ADC values of voxels from entire WM or GM regions, significant group differences were found for tNAA, tCr, and water ADC values in selected subregions such as the precuneus and cingulate cortex. This result likely reflects the spatial heterogeneity of microstructural and metabolic brain alterations in HIV infection [29,68]. Localized changes in ADC values may be masked when averaging over larger brain regions (i.e., between all GM and WM voxels), particularly in a small exploratory cohort with individual variability in disease progression and treatment history. Significant correlations were found at RACC between tNAA ADC and executive function, tCho ADC and GDS, and tCr ADC and memory (negative correlation). Significant correlations were also found between tCho ADC and depression at LACC (negative correlation), tCr ADC and EF at LPCR, and tCr ADC and anxiety at RSP, highlighting their potential as imaging biomarkers for cognitive and emotional impairment in PLWH [56,69]. These findings further support the clinical relevance of metabolite diffusion measures in monitoring disease progression and neurocognitive and neuroaffective status. Regions that are not reported in the results did not show any statistically significant differences.

Findings in the superior precuneus and corona radiata for tNAA, right anterior cingulate cortex for tCr, and cingulate cortex and corona radiata for water show that HIV infection is associated with region-specific increases in both water and metabolite ADCs. These changes reflect the pathophysiology of HIV in the brain, including glial activation and neuronal injury, and the effect of this pathology on function since the ADC values correlate with neurocognitive impairment in affected individuals [70,71].

Analysis of metabolite ratios revealed that the tNAA ratios were consistently reduced, while tCr and tCho ratios were elevated in PLWH compared to HC, with several regions showing statistically significant differences. These alterations in metabolite ratios align with the known pathophysiology of HIV-associated neurocognitive disorders, where neuronal loss (reflected by decreased NAA) and glial activation or inflammation (reflected by increased Cho and Cr) are reported [62,72,73,74,75].

The discriminant analysis demonstrated robust classification performance using both ADC and metabolite ratio metrics. The model incorporating water and tCr ADCs achieved high sensitivity (93.8%) and specificity (100%), with excellent predictive values and an AUC of 0.975, indicating strong potential for distinguishing PLWH from HC based on these imaging biomarkers. The second model, which included both tNAA ratio and tCr ADC, also showed good classification accuracy, albeit with slightly lower sensitivity and specificity than the first.

The correlations between ADC values and scores from these neuropsychological domains show potential associations between structural brain changes and cognitive and emotional functions [64,69]. For example, the moderate positive correlation of tNAA ADC at RACC with executive function, and significant correlations of tCr and tCho with memory, anxiety, and GDS, suggest that these imaging biomarkers may reflect underlying neuronal and glial processes contributing to cognitive and affective deficits in PLWH. Similarly, the observed pattern of lower tNAA and higher tCr and tCho ratios associated with HIV-infection supports their role as potential biomarkers for neurocognitive status.

Overall, these results underscore the utility of advanced MR spectroscopic imaging and diffusion metrics in detecting HIV-related brain changes and their association with cognitive function. The significant group differences, correlations with neuropsychological performance, and high discriminative power of selected biomarkers highlight their promise for clinical and research applications in HIV-associated neurocognitive disorder assessment and monitoring.

### Limitations

The small sample size and uneven gender distribution, with a predominance of males in the PLWH, reduce statistical power and limit the generalizability of the findings. The restricted availability of neuropsychological data in the HC group further limits the strength of cognitive correlations. Additionally, neuropsychological testing conducted via videoconferencing may have introduced some variability compared to traditional in-person assessment methods due to factors such as technical issues and participant familiarity with the technology. However, prior research supports videoconference-based testing as a generally valid and reliable alternative for many cognitive measures. The relatively long scan time (~45 min) for the diffusion spectroscopic imaging technique without acceleration could pose a challenge for clinical translation and raise the potential for motion artifacts affecting data quality. Furthermore, the complexity of HAND diagnosis, influenced by demographic, educational, and socioeconomic factors, suggests the need for cautious interpretation and standardization. Future research with larger, sex-balanced cohorts, more comprehensive neuropsychological evaluations, and optimized imaging protocols will be essential to validate and extend these preliminary findings.

While our LDA model demonstrated high classification accuracy and cross-validation performance, a small sample size remains an inherent limitation. Since the number of features used in the two LDA models (two and three features, respectively) is substantially lower than the number of samples per group, classical issues such as singular covariance matrices and highly unstable discriminant functions are largely mitigated. However, small datasets are inherently more vulnerable to sampling variability and may not fully capture population heterogeneity. Thus, while our LDA findings are promising, they should be interpreted cautiously and require validation in larger, independent cohorts. Future work incorporating additional samples and complementary classification approaches will strengthen the robustness and applicability of these models.

Further limitations include the following: (1) The study only assessed tNAA, tCr, and tCho due to the use of a long TE as a trade-off for achieving the single-shot trace-weighted ADC measurement, which limits the sensitivity to other metabolites like myo-Inositol, glutamine, and glutamate. (2) Neuropsychological data were not available for all HC, limiting the ability to perform comprehensive comparisons between PLWH and HC. Hence, this study primarily examined the correlation between spectroscopic imaging-based biomarkers (metabolite ADC and metabolite ratios) and neuropsychological data in PLWH, rather than the comparison of neuropsychological performance between PLWH and HC.

## 5. Conclusions

This study presents pilot results demonstrating that regional metabolite and water ADC values, as well as metabolite ratios, differ significantly between individuals with PLWH and HC, particularly in brain regions implicated in cognitive functions. Elevated ADC values in PLWH suggest increased diffusivity and underlying microstructural changes, while alterations in metabolite ratios reflect neuronal and glial pathology associated with HIV infection. The correlations between imaging biomarkers and neuropsychological performance, along with high classification accuracy achieved by discriminant analysis models, show the potential of these metrics for detecting and monitoring HIV-associated neurocognitive changes. These findings support the integration of diffusion and spectroscopic imaging measures into future clinical and research protocols aimed at early identification and management of neurocognitive impairment in PLWH.

## Figures and Tables

**Figure 1 metabolites-15-00669-f001:**
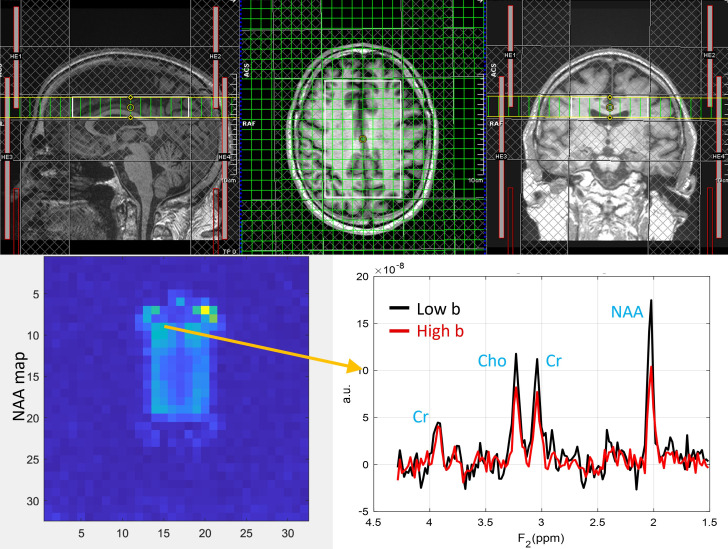
Localizer images (sagittal, axial, and coronal), the NAA metabolite map, and an extracted spectrum from a 63-year-old male subject with HIV (CD4 = 420 cells/µL, plasma HIV RNA < 20 copies/mL). The top row shows sagittal, axial, and coronal localizer images. The image at the bottom left shows the NAA metabolite map, and the bottom right shows an extracted spectrum. Low-b value (black) and high-b value (red) plots are overlaid.

**Figure 2 metabolites-15-00669-f002:**
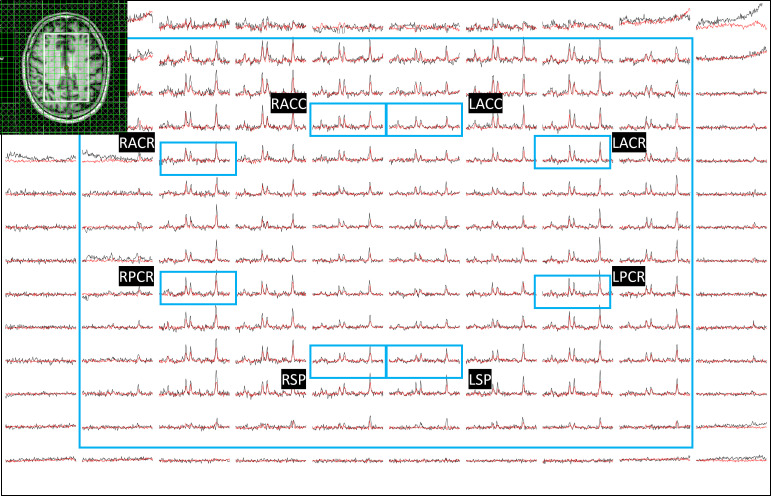
Multi-voxel plot from the VOI (white box in the axial localizer image) for the same subject in Figure 1 is depicted. Eight different regions used for the analysis are marked in green boxes. Right and left anterior cingulate cortex (RACC and LACC), right and left superior precuneus (RSP and LSP), and bilateral anterior and posterior corona radiata (RACR (right anterior corona radiata), RPCR (right posterior corona radiata), LACR (left anterior corona radiata)). Low-b value (black) and high-b value (red) plots are overlaid.

**Figure 3 metabolites-15-00669-f003:**
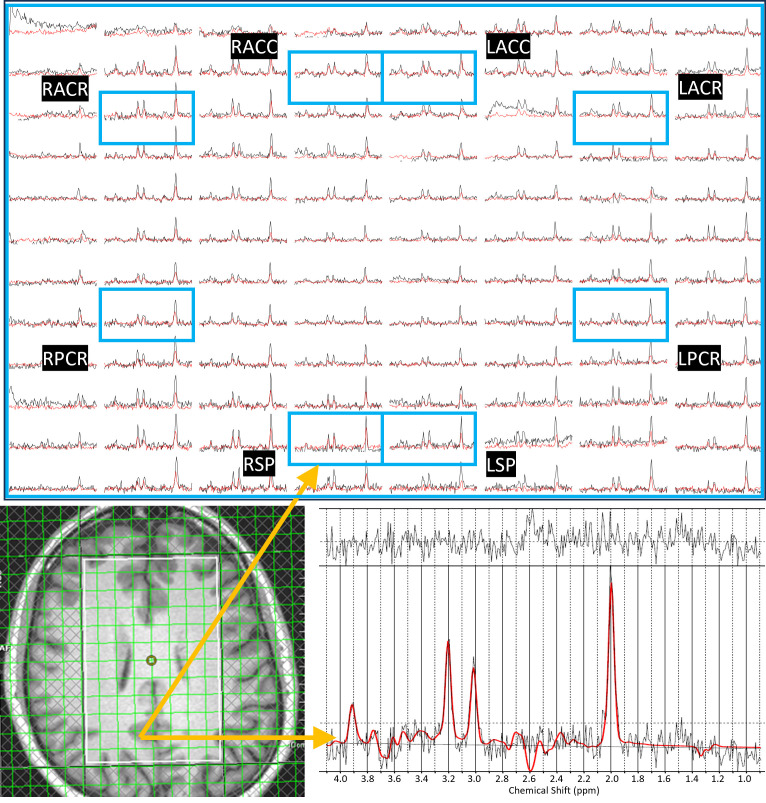
Axial localizer image and multivoxel spectra from a 38-year-old healthy control. Low-b value (black) and high-b value (red) plots are overlaid in the multivoxel spectra. The LCModel fit of an extracted spectrum is shown in the bottom right panel.

**Figure 4 metabolites-15-00669-f004:**
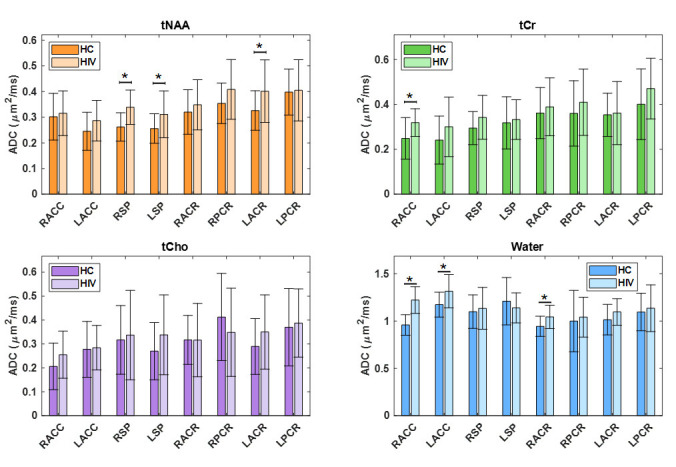
A bar chart comparing the ADC values of metabolites and water at different brain regions between HCl and PLWH. Significant differences (*p* < 0.05) are marked with an asterisk (*), while those that remain significant after Bonferroni correction are denoted with a double asterisk (**).

**Figure 5 metabolites-15-00669-f005:**
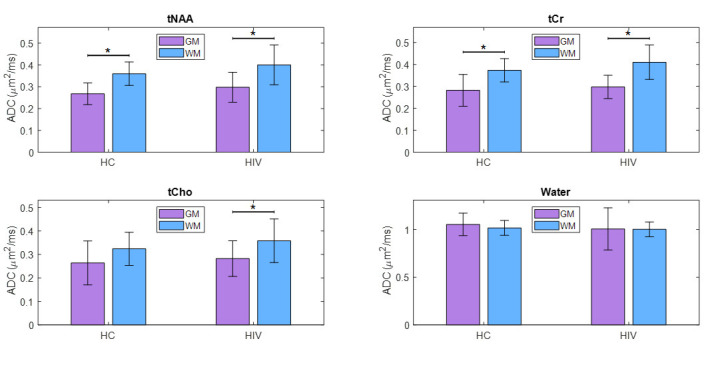
A bar chart comparing the extrapolated ADC values of gray matter (GM) and white matter (WM) between HC and PLWH. Significant differences (*p* < 0.05) are marked with an asterisk (*), while those that remain significant after Bonferroni correction are denoted with a double asterisk (**).

**Figure 6 metabolites-15-00669-f006:**
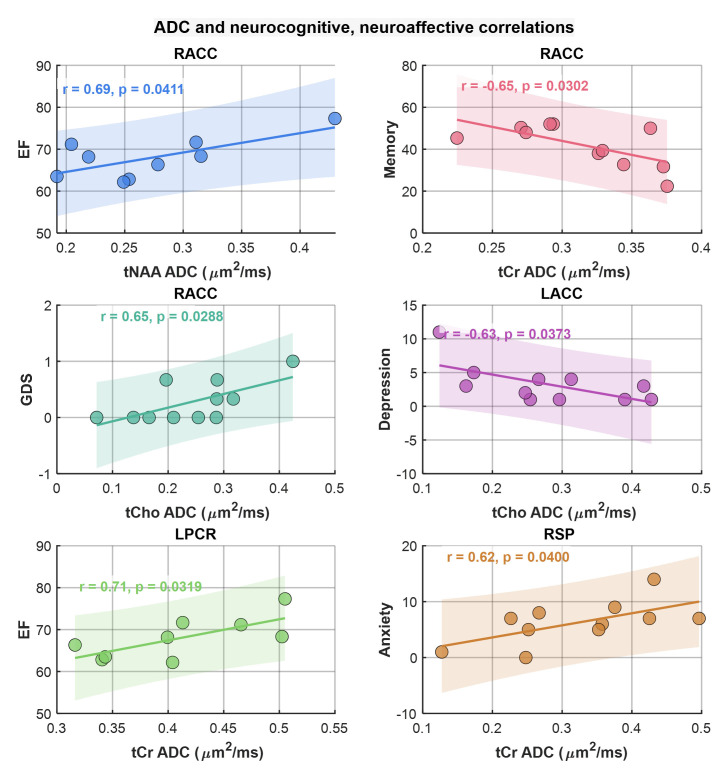
Scatter plots of metabolite ADCs showing significant correlations with neuropsychological domain scores. Pearson correlation (r) and *p*-values are shown in the inset.

**Figure 7 metabolites-15-00669-f007:**
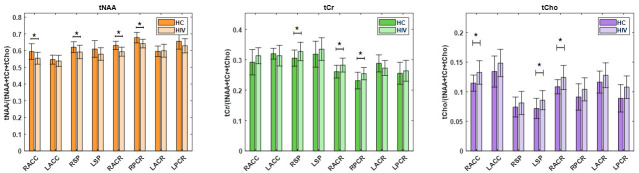
A bar chart comparing metabolite ratios across different brain regions between HC and PLWH. Significant differences (*p* < 0.05) are indicated with an asterisk (*), while differences that remain significant after Bonferroni correction are denoted with a double asterisk (**).

**Figure 8 metabolites-15-00669-f008:**
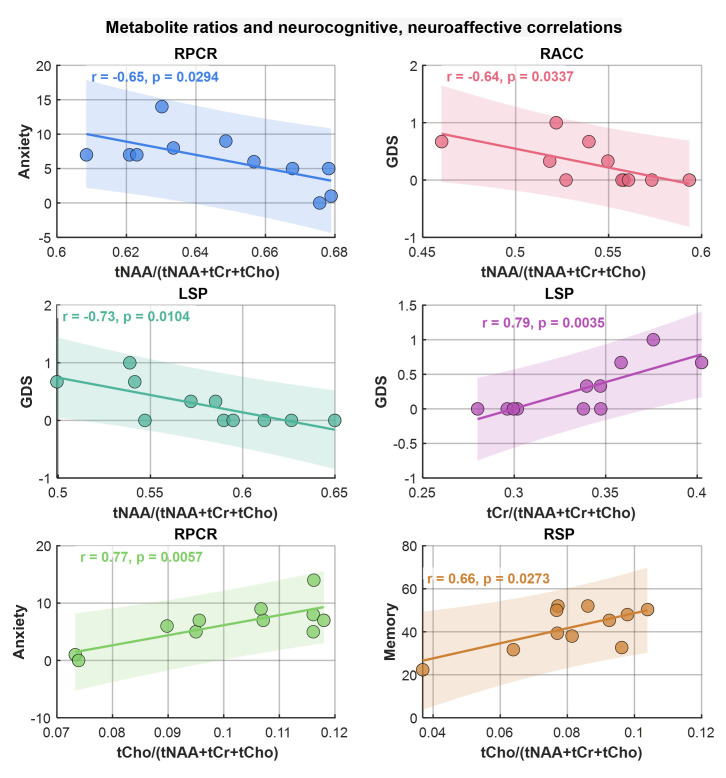
Scatter plots of metabolite ratios showing significant correlations with neuropsychological domain scores. Pearson correlation (r) and *p*-values are shown in the inset.

**Figure 9 metabolites-15-00669-f009:**
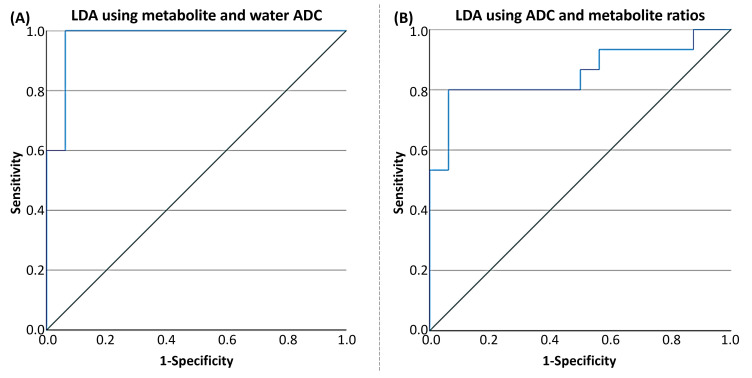
(**A**) ROC curve of the tCr + water ADC for differentiating healthy controls from HIV based on linear discriminant analysis (LDA), with an AUC of 0.975 (95% CI: 0.923–1.0). (**B**) ROC curve of the tNAA ratio + tCr ADC for differentiating HC from PLWH using LDA, with an AUC of 0.854 (95% CI: 0.708–1.0).

**Table 1 metabolites-15-00669-t001:** Baseline characteristics of study participants.

	HIV+	Healthy Controls	*p*-Value ^4^
N	16	15	N/A^2^
Age (Mean ± SD ^1^)	52.5 ± 10.5	37.6 ± 12.6	0.012
Gender (Male/Female)	15/1	12/3	0.33
CD4+ count cells/µL (Mean ± SD)	822.93 ± 319	N/A	N/A
CD4+ count > 200 cells/µL (%)	88 (*n* = 14)	N/A	N/A
viral count (HIV RNA PCR)	<20 (*n* = 9) >20 (*n* = 3) Undetectable (n = 4)	N/A	
Estimated duration of HIV infection (years, Mean ± SD)	24.18 ± 7.67	N/A	N/A
Percentage of years receiving cART ^3^	96	N/A	N/A

^1^ SD: standard deviation. ^2^ N/A: Not applicable or available. ^3^ cART: combination antiretroviral therapy. ^4^ Group comparison using Mann–Whitney U-test (Age) and Fischer’s Exact test (Gender).

## Data Availability

The datasets generated and/or analyzed during the current study are not publicly available due to ethical and data protection restrictions, but are available from the corresponding author on reasonable request and subject to an institutional data sharing agreement.

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
