# Peer review of "Association of Intracellular Microstructural and Neuropsychological Changes in HIV: A Pilot Validation of Trace Diffusion-Weighted Magnetic Resonance Spectroscopic Imaging Using Radial Trajectories"

_metabolites, 2025, doi:10.3390/metabo15100669_

Round 1
Reviewer 1 Report
Comments and Suggestions for Authors
ABSTRACT
- The small sample size and preliminary nature of the data should be explicitly stated.
- It should be noted that neuropsychological testing was conducted only in a limited subgroup of participants.
- The lack of statistically significant differences in metabolite and ADC values between groups has not been reported.
INTRODUCTION
- Updated epidemiological data on the prevalence of HAND (HIV-associated neurocognitive disorder) should be included.
- There is a contradiction in the stated study aims; objectives that assume group differences without supporting rationale are deemed unnecessary.
- The rationale for the use of metabolite and ADC measurements, and their relevance to the study aims, has not been sufficiently explained.
METHODS
- Demographic mismatch is present between groups: the HIV+ group is younger and predominantly male (Table 1). An ANCOVA controlling for age and sex should be conducted and reported.
- The validity of neuropsychological testing via videoconferencing is questionable. A clear protocol for remote administration should be included.
- There is uncertainty regarding LCModel output; Cramér-Rao Lower Bound (CRLB) thresholds for metabolite quantification (e.g., CRLB < 20%) must be stated.
- The selection of specific regions (e.g., precuneus, corona radiata) for radiological analysis should be justified with reference to HIV-related neuropathology.
RESULTS
- A contradiction exists between the statement that there is “no difference in metabolite ADCs between HIV and HC groups” (Figure 5) and “regional differences exist” (Figure 4). This inconsistency is not addressed in the discussion.
- Multiple comparison error is present (8 regions × 4 metabolites = 32 comparisons); correction methods such as Bonferroni or FDR should be applied (e.g., adjusted p < 0.0016).
- Since neuropsychological data were collected in only 6 controls, correlations should be restricted to the HIV+ group. Current analyses inappropriately include all participants.
- The reported 100% specificity with a sample size of N=31 suggests overfitting; bootstrap-based cross-validation should be included.
DISCUSSION
- Sample size limitations, demographic imbalances, and the restricted availability of neuropsychological data should be the first limitations discussed, and elaborated upon in detail.
- The inconsistency between the absence of group differences in metabolite ADCs (Figure 5) and the presence of regional differences (Figure 4) needs to be addressed.
- The scan duration is not practical for clinical application; this should be emphasized as a pilot technical approach.
REFERENCES
- The reference list is outdated and lacks coverage of recent neuroimaging studies in HIV populations.
GENERAL RECOMMENDATIONS
- The manuscript contains several typographical errors and figure inconsistencies (e.g., “p < 0.5” in Figure 5 should be corrected to “p < 0.05”; AUC = 0.854 with CI [0.708–1] in Figure 9B is invalid—confidence intervals cannot exceed 1).
- The limitations of LDA findings due to small sample size have not been sufficiently discussed in the manuscript.
Although the article addresses a scientifically important topic of potential relevance to the field, it is currently considered inadequate for publication due to small sample size, methodological deficiencies, and limited robustness in statistical analysis.
Author Response
Reviewer #1:
Abstract
- Comment: The small sample size and preliminary nature of the data should be explicitly stated.
- Response: Thank you for this suggestion. We have updated the abstract to explicitly mention the small sample size and the preliminary nature of our data to provide clear context for the study’s limitations.
Last sentence in Abstract: “However, the small sample size and preliminary nature of this data warrant further investigation to validate these findings.”
- Comment: It should be noted that neuropsychological testing was conducted only in a limited subgroup of participants.
- Response: We have revised the abstract to include a statement highlighting that neuropsychological testing was only conducted in the HIV group and the correlation was performed only within the HIV group.
“Neuropsychological testing was conducted only in the HIV group, and the correlation between ADC values and neuropsychological domain scores was performed exclusively within this group.”
- Comment: The lack of statistically significant differences in metabolite and ADC values between groups has not been reported.
- Response: Thank you for the comment. In this study, we reported significant differences in metabolite ratios and metabolite ADC values between the HIV+ and HC groups. We have now added the following statement in the discussion section of the revised manuscript.
“Regions which are not reported in the results did not show any statistically significant differences.”.
Introduction
- Comment: Updated epidemiological data on the prevalence of HAND should be included.
- Response: We have included recent epidemiological data on the prevalence of HAND as follows.
“HAND continues to affect a substantial proportion of people living with HIV (PLWH), posing significant challenges despite advances in antiretroviral therapy (ART). Recent cohort studies estimate the global prevalence of HAND to range from approximately 25% to 60%, with variation due to differences in diagnostic criteria, study populations, and geographic settings (1). The most common subtype, asymptomatic neurocognitive impairment (ANI), accounts for about 28%, followed by mild neurocognitive disorder (MND), while HIV-associated dementia (HAD) has become relatively rare in the ART era (2). Although effective ART has significantly reduced the incidence of HAD, the overall prevalence of HAND has remained stable or even increased in some populations due to prolonged survival and aging of PLWH (3). Risk factors contributing to HAND include older age, lower education, comorbid conditions, and duration of HIV infection (4). HAND is associated with impairments in executive function, memory, and attention that reduce quality of life and treatment adherence. Consequently, ongoing research to identify sensitive biomarkers and improve diagnosis remains vital to address this persistent neurological complication in HIV (5).”
- Zenebe Y, Necho M, Yimam W, Akele B. Worldwide occurrence of HIV-associated neurocognitive disorders and its associated factors: a systematic review and meta-analysis. Frontiers in psychiatry. 2022 May 31;13:814362.
- Zhou Z, Wang W, Li H, Shi Y, Zhao L, Lu Y, Wei X, Li H. Decoding HIV-associated neurocognitive disorders: a new perspective from multimodal connectomics. Frontiers in Neurology. 2025 Jan 29;16:1467175.
- Mastrorosa I, Pinnetti C, Brita AC, Mondi A, Lorenzini P, Del Duca G, Vergori A, Mazzotta V, Gagliardini R, Camici M, De Zottis F. Declining prevalence of human immunodeficiency virus (HIV)–associated neurocognitive disorders in recent years and associated factors in a large cohort of antiretroviral therapy–treated individuals with HIV. Clinical Infectious Diseases. 2023 Feb 1;76(3):e629-37.
- Kelebie MA, Tinsae T, Alemayehu BF, Walelign GK, Takelle GM. Prevalence and associated factors of neurocognitive disorder among people living with HIV/AIDS in the South Gondar zone primary hospitals, North-West Ethiopia: an institution-based cross-sectional study. BMJ open. 2024 May 1;14(5):e082773.
- Saylor D, Dickens AM, Sacktor N, Haughey N, Slusher B, Pletnikov M, Mankowski JL, Brown A, Volsky DJ, McArthur JC. HIV-associated neurocognitive disorder—pathogenesis and prospects for treatment. Nature Reviews Neurology. 2016 Apr;12(4):234-48.
- Comment: There is a contradiction in the stated study aims; objectives that assume group differences without supporting rationale are deemed unnecessary.
- Response: We have the following paragraphs in the Introduction to ensure that our objectives are clearly justified with supporting rationale.
“Magnetic resonance spectroscopy (MRS) metabolite measurements and apparent diffusion coefficient (ADC) assessments from DWI provide complementary information on brain tissue biochemical and microstructural properties affected by HIV infection. Key metabolites quantified by MRS include total N-acetylaspartate (tNAA), an established marker of neuronal integrity; total creatine (tCr), involved in cellular energy metabolism; and total choline (tCho), reflecting membrane turnover and inflammatory activity. Prior research has demonstrated alterations in these metabolites in HIV-infected individuals, consistent with neuroinflammatory processes, glial activation, and neuronal injury (1,2). The ADC metric captures the diffusivity of metabolites and water within tissues, which can be altered by microstructural changes such as cellular swelling, loss of tissue barriers, or inflammatory edema. For example, increased ADC values of creatine have been specifically reported in conditions of hypermetabolism and neuroinflammation, suggesting altered energy metabolism and microglial activation underlying neuropathology (3,4). Combining MRS metabolic profiling with diffusion-based ADC measures thus allows a more sensitive and comprehensive characterization of HIV-associated brain changes, potentially serving as early biomarkers for neurocognitive impairment and monitoring treatment effects.”
- Chelala L, O'Connor EE, Barker PB, Zeffiro TA. Meta-analysis of brain metabolite differences in HIV infection. NeuroImage: Clinical. 2020 Jan 1;28:102436.
- Chaganti J, Brew BJ. MR spectroscopy in HIV associated neurocognitive disorder in the era of cART: a review. AIDS Research and Therapy. 2021 Oct 9;18(1):65.
- Marques D, Carecho R, Carregosa D, dos Santos CN. The Potential of Low Molecular Weight (Poly) phenol Metabolites for Attenuating Neuroinflammation and Treatment of Neurodegenerative Diseases. Recent Advances in Polyphenol Research. 2023 Mar 3;8:95-138.
- Mudra Rakshasa-Loots A, Diteko G, Dowell NG, Ronen I, Vera JH. Neuroimmunometabolic alterations and severity of depressive symptoms in people with HIV: An exploratory diffusion-weighted MRS study. Brain and Neuroscience Advances. 2025 Apr;9:23982128251335792.
- Comment: The rationale for the use of metabolite and ADC measurements, and their relevance to the study aims, has not been sufficiently explained.
- Response: The rationale for using metabolite and ADC measurements has been expanded in the Introduction as mentioned in the response to previous comment.
Methods
- Comment: Demographic mismatch is present between groups: the HIV+ group is younger and predominantly male (Table 1). An ANCOVA controlling for age and sex should be conducted and reported.
- Response: Thank you for pointing this out. Due to the small sample size and uneven gender distribution, particularly the predominance of males in the HIV+ group, conducting an ANCOVA controlling for age and sex was not feasible without risking unreliable or biased results. Instead, a Mann-Whitney U test was employed to compare age distributions between HIV+ and healthy control groups (1). The test showed no significant difference in age (p < 0.05), supporting comparability of groups on this measure. Furthermore, to minimize confounding effects from uneven sex distribution (2), the single female participant in the HIV+ group and three female participants from healthy control group were excluded from analyses. We had already excluded the single female participant from the correlation analysis presented in initial submission. This is now explicitly stated in the revised manuscript. We have also noticed that the information in Table 1 had errors/incomplete in the initial submission. This is corrected in the updated manuscript.
- Day RW, Quinn GP. Comparisons of treatments after an analysis of variance in ecology. Ecological monographs. 1989 Dec;59(4):433-63.
- Janušonis et al., 2009. Comparing statistical methods for small neuroscience samples. Journal of Neuroscience Methods.
- Comment: The validity of neuropsychological testing via videoconferencing is questionable. A clear protocol for remote administration should be included.
- Response: Thank you for the comment. We have added the following references to early research reports that have shown that remote assessment with a variety of neuropsychological tests, including those used in this study, are reliable, valid, and well tolerated by participants. We have also mentioned in the limitations that “neuropsychological testing was conducted via videoconferencing, which may have introduced variability compared to in-person assessment methods”
- Rizzi E, Vezzoli M, Pegoraro S, Facchin A, Strina V, Daini R. Teleneuropsychology: Normative data for the assessment of memory in online settings. Neurological Sciences. 2023 Feb;44(2):529-38.
- Brown T, Zakzanis KK. A review of the reliability of remote neuropsychological assessment. Applied Neuropsychology: Adult. 2025 Sep 3;32(5):1536-42.
- Wärn E, Andersson L, Berginström N. Remote Neuropsychological Testing as an Alternative to Traditional Methods—a Convergent Validity Study. Archives of Clinical Neuropsychology. 2025 Feb 20:acaf013.
- Comment: There is uncertainty regarding LCModel output; Cramér-Rao Lower Bound (CRLB) thresholds for metabolite quantification (e.g., CRLB < 20%) must be stated.
- Response: We have clarified the LCModel output and have explicitly stated the CRLB thresholds used for metabolite quantification (CRLB < 20%) in the Methods section.
- Comment: The selection of specific regions (e.g., precuneus, corona radiata) for radiological analysis should be justified with reference to HIV-related neuropathology.
- Response: We have included the following justification for the selection of specific regions (precuneus, corona radiata) in the radiological analysis, referencing relevant studies on HIV-related neuropathology, in section 2.4 of the revised manuscript.
“The regions analyzed in this study include right and left anterior cingulate cortex (RACC, LACC), right and left superior precuneus (RSP, LSP), and bilateral anterior and posterior corona radiata (RACR, RPCR, LACR, LPCR. These regions have well-established involvement in HIV-associated neuropathology and cognitive dysfunction. The anterior cingulate cortex (ACC) is crucial for executive function, attention, and emotional regulation, with numerous studies linking ACC structural and functional alterations to HIV-related cognitive impairment (1,2). The precuneus is implicated in memory, visuospatial processing, and self-referential thought and is frequently affected in HIV, with neuroimaging consistently showing metabolic and connectivity disruptions correlated with cognitive deficits (3). The corona radiata, comprising essential white matter tracts, facilitates cortical-subcortical communication vital for executive functions and the diffusion abnormalities in these regions are strongly associated with HAND and cognitive decline (4).”
- Xu F, Ma J, Wang W, Li H. A longitudinal study of the brain structure network changes in HIV patients with ANI: combined VBM with SCN. Frontiers in neurology. 2024 Apr 17;15:1388616.
- Zhou Z, Wang W, Li H, Shi Y, Zhao L, Lu Y, Wei X, Li H. Decoding HIV-associated neurocognitive disorders: a new perspective from multimodal connectomics. Frontiers in Neurology. 2025 Jan 29;16:1467175.
- Zhou Z, Gong W, Hu H, Wang F, Li H, Xu F, Li H, Wang W. Functional and Structural Network Alterations in HIV-Associated Asymptomatic Neurocognitive Disorders: Evidence for Functional Disruptions Preceding Structural Changes. Neuropsychiatric Disease and Treatment. 2025 Dec 31:689-709.
- Leite SC, Corrêa DG, Doring TM, Kubo TT, Netto TM, Ferracini R, Ventura N, Bahia PR, Gasparetto EL. Diffusion tensor MRI evaluation of the corona radiata, cingulate gyri, and corpus callosum in HIV patients. Journal of Magnetic Resonance Imaging. 2013 Dec;38(6):1488-93.
Results
- Comment: A contradiction exists between the statement that there is “no difference in metabolite ADCs between HIV and HC groups” (Figure 5) and “regional differences exist” (Figure 4). This inconsistency is not addressed in the discussion.
- Response: We have clarified this statement in the discussion section. Although no significant group differences were observed in the extrapolated ADC values, regional differences were identified, which we attribute to the exploratory nature of the study and variability in specific regions of interest.
The following is added in the revised manuscript section 4: “Although no significant differences were observed in the extrapolated metabolite ADC values of voxels from entire WM or GM regions, significant group differences were found for tNAA, tCr, and water ADC values in selected subregions such as the precuneus and cingulate cortex. This result likely reflects the spatial heterogeneity of microstructural and metabolic brain alterations in HIV infection (1, 2). Localized changes in ADC val-ues may be masked when averaging over larger brain regions (i.e, between all GM and WM voxels), particularly in a small exploratory cohort with individual variability in dis-ease progression and treatment history.”
- Chelala L, O'Connor EE, Barker PB, Zeffiro TA. Meta-analysis of brain metabolite differences in HIV infection. NeuroImage: Clinical. 2020 Jan 1;28:102436.
- Hua X, Boyle CP, Harezlak J, Tate DF, Yiannoutsos CT, Cohen R, Schifitto G, Gongvatana A, Zhong J, Zhu T, Taylor MJ. Disrupted cerebral metabolite levels and lower nadir CD4+ counts are linked to brain volume deficits in 210 HIV-infected patients on stable treatment. NeuroImage: clinical. 2013 Jan 1;3:132-42.
- Comment: Multiple comparison error is present (8 regions × 4 metabolites = 32 comparisons); correction methods such as Bonferroni or FDR should be applied (e.g., adjusted p < 0.0016).
- Response: Since the analysis in 8 regions were independent, we have added Bonferroni correction for multiple comparison (4 metabolites) and set the p value threshold at p < 0.0125. We have now applied the correction and updated the results accordingly.
- Comment: Since neuropsychological data were collected in only 6 controls, correlations should be restricted to the HIV+ group. Current analyses inappropriately include all participants.
- Response: Thank you for your comment. However, we would like to clarify that in the original submission, we already restricted the correlation analyses to the HIV+ group, as the small sample size of controls makes such comparisons less meaningful. The analyses were conducted within the HIV+ group to ensure a more robust and interpretable result.
- Comment: The reported 100% specificity with a sample size of N=31 suggests overfitting; bootstrap-based cross-validation should be included.
- Response: We acknowledge the concern regarding overfitting. We have shown the cross-validated results in the revised manuscript.
Discussion
- Comment: Sample size limitations, demographic imbalances, and the restricted availability of neuropsychological data should be the first limitations discussed, and elaborated upon in detail.
- Response: We have modified the limitation sub-section in the discussion section to address the limitations of sample size, demographic imbalances, and restricted neuropsychological data in more detail.
- Comment: The inconsistency between the absence of group differences in metabolite ADCs (Figure 5) and the presence of regional differences (Figure 4) needs to be addressed.
- Response: Thank you for the comment. We explain that regional differences may reflect localized neurobiological changes, even in the absence of group-wide differences.
- Comment: The scan duration is not practical for clinical application; this should be emphasized as a pilot technical approach.
- Response: We have added a statement highlighting that the scan duration is a limitation for clinical translation and emphasize that this study represents a pilot investigation into technical feasibility. Our future work is focused on further acceleration to reduce the scan time to 15-30 minutes.
References
- Comment: The reference list is outdated and lacks coverage of recent neuroimaging studies in HIV populations.
- Response: We have updated the reference list to include recent studies on neuroimaging in HIV populations, ensuring that the manuscript reflects current research in the field.
General Recommendations
- Comment: The manuscript contains several typographical errors and figure inconsistencies (e.g., “p < 0.5” in Figure 5 should be corrected to “p < 0.05”; AUC = 0.854 with CI [0.708–1] in Figure 9B is invalid—confidence intervals cannot exceed 1).
- Response: We have thoroughly proofread the manuscript and corrected typographical errors, as well as figure inconsistencies (e.g., correcting “p < 0.5” to “p < 0.05” and addressing the invalid CI in Figure 9B).
- Comment: The limitations of LDA findings due to small sample size have not been sufficiently discussed in the manuscript.
- Response: Thanks for the comment. We have added the following in the limitation section.
“While our LDA model demonstrated high classification accuracy and cross-validation performance, small sample size remains an inherent limitation. Since the number of features used in the two LDA models (2 and 3 features, respectively) is substantially lower than the number of samples per group, classical issues such as singular covariance matrices and highly unstable discriminant functions are largely mitigated. However, small datasets are inherently more vulnerable to sampling variability and may not fully capture population heterogeneity. Thus, while our LDA findings are promising, they should be interpreted cautiously and require validation in larger, independent cohorts. Future work incorporating additional samples and complementary classification approaches will strengthen the robustness and applicability of these models.”
Reviewer 2 Report
Comments and Suggestions for Authors
Dear Authors, see comments below.
- The paper presents a large set of experimental data. Figures 1-8 are shown in a row without preliminary intermediate descriptions, which complicates the perception of information.
- In Fig. 1. from the bottom right is shown a comparison of two spectra. Judging by the superposition of the leftmost signal, the change in the relative intensity of the remaining components is significant. Low b has a higher noise level than High b, so the relative comparison must be done with caution. It is also necessary to provide an explanation of the identification of each line.
- The histograms are shown with a large confidence interval, where a small difference between different columns compared to Error Bars reduces the statistical power of the comparative analysis.
- Figure 9 has an extremely small discretization step, but in general for a qualitative analysis this may be sufficient.
- There are typos throughout the text that can be easily corrected.
Author Response
Reviewer #2:
- Comment: The paper presents a large set of experimental data. Figures 1-8 are shown in a row without preliminary intermediate descriptions, which complicates the perception of information.
- Response: We have reorganized Figures 1-8 by grouping them more logically to improve clarity and facilitate interpretation.
- Comment: In Fig. 1, from the bottom right is shown a comparison of two spectra. Judging by the superposition of the leftmost signal, the change in the relative intensity of the remaining components is significant. Low b has a higher noise level than High b, so the relative comparison must be done with caution. It is also necessary to provide an explanation of the identification of each line.
- Response: Thank you for the comment. The difference in noise level is due to the lower number of averages used in the acquisition for low-b value. This is now explicitly stated in the revised manuscript.
- Comment: The histograms are shown with a large confidence interval, where a small difference between different columns compared to error bars reduces the statistical power of the comparative analysis.
- Response: Thank you for the comment. As the reviewer pointed out, the histograms with larger confidence intervals reflect higher variability within the data, which corresponds to the absence of statistically significant differences. The bar plots with significant differences are already marked with an asterisk, as indicated in the figure. This distinction helps to clearly highlight the points where statistical significance was achieved.
- Comment: Figure 9 has an extremely small discretization step, but in general for a qualitative analysis this may be sufficient.
- Response: Thank you for your comment and for pointing out that the small discretization step used for the ROC curve is indeed sufficient for qualitative analysis.
- Comment: There are typos throughout the text that can be easily corrected.
- Response: We have thoroughly proofread the manuscript and corrected all typographical errors.
Reviewer 3 Report
Comments and Suggestions for Authors
Dear Authors,
I have reviewed your article entitled “Association Of Intracellular Microstructural and Neuro-psychological Changes in HIV: A Pilot Validation of Trace Diffusion-Weighted Magnetic Resonance Spectroscopic Imaging using Radial trajectories”
Thank you for your efforts. Overall, it's well designed and written.
You should just make a few revisions:
- Table 1 and related information should be moved in the results section.
- The date of the study is not written. It should be added.
- HIV-RNA levels were not seen in Table 1.
- The number of patients is small, but I wonder if these values ​​vary depending on the duration of the HIV? Could they be divided into groups based on the duration of the HIV?
- The abbreviation (RPCP: right posterior corona radiata) is misspelled on line 244.
Best regards,
Author Response
Reviewer #3:
- Comment: Table 1 and related information should be moved to the results section.
- Response: We have moved Table 1 and related information to the Results section, as suggested.
- Comment: The date of the study is not written. It should be added.
- Response: Thank you for pointing this out. We have added the date of the study to the Methods section of the revised document. 2021-2024
- Comment: HIV-RNA levels were not seen in Table 1.
- Response: Thank you for pointing this out. We have now included the HIV-RNA viral load values in table 1.
- Comment: The number of patients is small, but I wonder if these values vary depending on the duration of HIV? Could they be divided into groups based on the duration of HIV?
- Response: We appreciate this suggestion. Given the small sample size, we were not able to explore the potential impact of HIV duration by dividing the HIV+ group into subgroups based on disease duration. We will consider this approach in future studies with a larger patient population.
- Comment: The abbreviation (RPCP: right posterior corona radiata) is misspelled on line 244.
- Response: We have corrected this in the revised manuscript.
Reviewer 4 Report
Comments and Suggestions for Authors
Thank you for providing opportunity to review the article
The small and uneven sample size substantially limits statistical power and the generalizability of findings. Authors should acknowledge this limitation more explicitly in the Discussion and avoid overinterpretation of results.
Please clarify- Since full cognitive testing was not performed in all HC, comparisons in cognitive domains are incomplete. This limits the ability to attribute cognitive–metabolite correlations specifically to HIV-related pathology.
Cognitive testing was conducted via video teleconferencing for some participants. Remote testing may introduce variability compared to in-person assessment, and its reliability in PLWH should be discussed.
The reduced number of HC with neuropsychological data limits between-group comparisons, making the correlation analyses primarily within PLWH. This should be clarified.
The advanced DW-REPSI sequence has a total scan time of ~45 minutes, which is long for clinical translation. Motion sensitivity during such long scans could influence ADC estimates. There is no mention of whether motion correction beyond self-navigation was used or how motion artifacts were handled.
Author Response
Reviewer #4:
- Comment: The small and uneven sample size substantially limits statistical power and the generalizability of findings. Authors should acknowledge this limitation more explicitly in the Discussion and avoid overinterpretation of results.
- Response: Thank you for pointing this out. We have now included a more detailed discussion about the limitations of the study in the discussion that specifically mentions the limitations in terms of statistical power and the generalizability of findings.
- Comment: Please clarify—since full cognitive testing was not performed in all HC, comparisons in cognitive domains are incomplete. This limits the ability to attribute cognitive–metabolite correlations specifically to HIV-related pathology.
- Response: Thank you for the comment. Full cognitive testing was only conducted in HIV, and as hence, cognitive-metabolite correlations are discussed only in the HIV+ group. We have added the following paragraph in section 2.4 of the revised manuscript.
“Since neuropsychological data were collected for only six healthy controls, we restricted correlation analyses to the HIV-positive group to avoid biased or uninterpretable findings due to the small control sample. This approach aligns with best practices in HIV neurocognitive research, where limited control data necessitates focusing on correlation assessments within HIV cohorts [1,2].”
- Gelman BB et al., Neurovirological correlation with HIV-associated neurocognitive impairment, J Neurovirol, 2013.
- Robertson K, Liner J, Heaton R. Neuropsychological assessment of HIV-infected populations in international settings. Neuropsychology review. 2009 Jun;19(2):232-49.
- Comment: Cognitive testing was conducted via video teleconferencing for some participants. Remote testing may introduce variability compared to in-person assessment, and its reliability in PLWH should be discussed.
- Response: Thank you for the comment. We have added the following references to early research reports that have shown that remote assessment with a variety of neuropsychological tests, including those used in this study, are reliable, valid, and well tolerated by participants. We have also mentioned in the limitations that “neuropsychological testing was conducted via videoconferencing, which may have introduced variability compared to in-person assessment methods”
- Rizzi E, Vezzoli M, Pegoraro S, Facchin A, Strina V, Daini R. Teleneuropsychology: Normative data for the assessment of memory in online settings. Neurological Sciences. 2023 Feb;44(2):529-38.
- Brown T, Zakzanis KK. A review of the reliability of remote neuropsychological assessment. Applied Neuropsychology: Adult. 2025 Sep 3;32(5):1536-42.
- Wärn E, Andersson L, Berginström N. Remote Neuropsychological Testing as an Alternative to Traditional Methods—a Convergent Validity Study. Archives of Clinical Neuropsychology. 2025 Feb 20:acaf013.
- Comment: The reduced number of HC with neuropsychological data limits between-group comparisons, making the correlation analyses primarily within PLWH. This should be clarified.
- Response: Thank you for the comment. This is now explicitly mentioned in the methods and discussion sections of the revised manuscripts.
- Comment: The advanced DW-REPSI sequence has a total scan time of ~45 minutes, which is long for clinical translation. Motion sensitivity during such long scans could influence ADC estimates. There is no mention of whether motion correction beyond self-navigation was used or how motion artifacts were handled.
- Response: We have acknowledged in the revised manuscript that the 45-minute scan duration is a limitation for clinical translation. We have also clarified that motion correction beyond self-navigation was not performed in this study, but this will be a critical consideration for future work.
Round 2
Reviewer 1 Report
Comments and Suggestions for Authors
The study was deemed to lack sufficient sample size and methodological quality to warrant publication.